# Identification of an Immunogenic Medulloblastoma-Specific Fusion Involving *EPC2* and *GULP1*

**DOI:** 10.3390/cancers13225838

**Published:** 2021-11-21

**Authors:** Claudia Paret, Nadine Lehmann, Hannah Bender, Maximilian Sprang, Clemens J. Sommer, Denis Cana, Larissa Seidmann, Arthur Wingerter, Marie A. Neu, Khalifa El Malki, Francesca Alt, Lea Roth, Federico Marini, Malte Ottenhausen, Martin Glaser, Markus Knuf, Alexandra Russo, Joerg Faber

**Affiliations:** 1Center for Pediatric and Adolescent Medicine, Department of Pediatric Hematology/Oncology, University Medical Center of the Johannes Gutenberg-University Mainz, 55131 Mainz, Germany; Nadine.Lehmann@unimedizin-mainz.de (N.L.); Bender.hannah@gmail.com (H.B.); masprang@uni-mainz.de (M.S.); Arthur.Wingerter@unimedizin-mainz.de (A.W.); Marie.Neu@unimedizin-mainz.de (M.A.N.); Khalifa.ElMalki@unimedizin-mainz.de (K.E.M.); Francesca.Alt@unimedizin-mainz.de (F.A.); Lea.Roth@unimedizin-mainz.de (L.R.); Alexandra.Russo@unimedizin-mainz.de (A.R.); Joerg.Faber@unimedizin-mainz.de (J.F.); 2University Cancer Center (UCT), University Medical Center of the Johannes Gutenberg-University Mainz, 55131 Mainz, Germany; 3German Cancer Consortium (DKTK), Site Frankfurt/Mainz, Germany, German Cancer Research Center (DKFZ), 69120 Heidelberg, Germany; 4Institute of Neuropathology, University Medical Center of the Johannes Gutenberg-University Mainz, 55131 Mainz, Germany; Clemens.Sommer@unimedizin-mainz.de (C.J.S.); Denis_Cana@klinikum-hanau.de (D.C.); 5Institute of Pathology, University Medical Center of the Johannes Gutenberg-University Mainz, 55131 Mainz, Germany; Larissa.Seidmann@unimedizin-mainz.de; 6Institute of Medical Biostatistics, Epidemiology and Informatics (IMBEI), University Medical Center of the Johannes Gutenberg-University Mainz, 55131 Mainz, Germany; marinif@uni-mainz.de; 7Department of Neurosurgery, University Medical Center of the Johannes Gutenberg-University Mainz, 55131 Mainz, Germany; Malte.Ottenhausen@unimedizin-mainz.de (M.O.); Martin.Glaser@unimedizin-mainz.de (M.G.); 8Children’s Hospital Worms, 67550 Worms, Germany; Markus.Knuf@klinikum-worms.de

**Keywords:** medulloblastoma, EPC2, GULP1, fusion

## Abstract

**Simple Summary:**

Medulloblastoma is the most common malignant childhood brain tumor and it is considered poor immunogenic because of its low mutational burden. Nevertheless, several clinical trials are currently evaluating immunotherapy for medulloblastoma patients, since new treatment strategies for this entity are a matter of utmost urgency. Tumor specific antigens resulting from gene fusions are potentially highly immunogenic. In our study, we identified a new medulloblastoma-specific fusion transcript *EPC2-GULP1*.The resulting protein sequence produced a neoantigen, which was able to activate CD8^+^ T cells. Thus, our data indicate an immunotherapeutic approach for pediatric medulloblastoma patients carrying the *EPC2-GULP1* fusion or other fusions generating immunogenic neoantigens.

**Abstract:**

Medulloblastoma is the most common malignant brain tumor in children. Immunotherapy is yet to demonstrate dramatic results in medulloblastoma, one reason being the low rate of mutations creating new antigens in this entity. In tumors with low mutational burden, gene fusions may represent a source of tumor-specific neoantigens. Here, we reviewed the landscape of fusions in medulloblastoma and analyzed their predicted immunogenicity. Furthermore, we described a new in-frame fusion protein identified by RNA-Seq. The fusion involved two genes on chromosome 2 coding for the enhancer of polycomb homolog 2 (EPC2) and GULP PTB domain containing engulfment adaptor 1 (GULP1) respectively. By qRT-PCR analysis, the fusion was detected in 3 out of 11 medulloblastoma samples, whereby 2 samples were from the same patients obtained at 2 different time points (initial diagnosis and relapse), but not in other pediatric brain tumor entities. Cloning of the full-length sequence indicated that the fusion protein contains the N-terminal enhancer of polycomb-like domain A (EPcA) of EPC2 and the coiled-coil domain of GULP1. In silico analyses predicted binding of the neoantigen-derived peptide to HLA-A*0201. A total of 50% of the fusions described in the literature were also predicted to produce an immunogenic peptide. The EPC2-GULP1 fusion peptide was able to induce a de novo T cell response characterized by interferon gamma release of CD8^+^ cytotoxic T cells in vitro. While the functional relevance of this fusion in medulloblastoma biology remains to be clarified, our data support an immunotherapeutic approach for pediatric medulloblastoma patients carrying the *EPC2-GULP1* fusion and other immunogenic fusions.

## 1. Introduction

Medulloblastoma (MB) is the most common malignant brain tumor in children. It accounts for approximately 20% of all pediatric central nervous system tumors. Histopathological classifications differentiate between five subtypes: classic, desmoplastic/nodular, medulloblastoma with extensive nodularity, anaplastic, and large cell medulloblastoma [1]. This classification system is complemented by a molecular classification system, which distinguishes between four subtypes depending on involved molecular pathways: WNT, SHH, group 3, and group 4 [2,3].

Current treatment protocols are based on risk stratification (standard-risk and high-risk for recurrence) and involve multimodal therapeutic approaches (surgery, craniospinal radiation, chemotherapy). These treatment strategies have shown an improvement in 5-year overall survival to 85% for children with standard-risk disease and ~ 70% for those with high-risk disease [1]. However, long-term survival is often associated with treatment-related morbidity, and late relapses are still possible, particularly in adult medulloblastoma. Thus, there is a critical need for more effective therapies to combat this disease.

Immunotherapy is revolutionizing cancer care, also for brain tumors patients. Activated T cells can be primed against tumor-specific antigens and traverse the blood brain barrier (BBB) through adhesion markers (i.e., VLA-4), allowing them to penetrate the tumor microenvironment and induce their effector functions against cancer cells [4,5]. To induce an immune response against tumors, appropriate tumor antigens need to be selected and targeted [6].

Fusion proteins resulting from chromosomal translocations in tumors can create neoantigens at the breakpoint, which are unique to the tumor cells. Since the first description of a translocation in cancer by Rowley in 1973 [7], translocations have been an important focus in cancer research. Translocations have been used not only as markers for certain cancer types, but also as therapeutic targets, since the fusion transcript may include druggable structures such as protein kinases [8]. Therefore, genome sequences of neoplastic cells are frequently analyzed for translocations to find new potential therapeutic targets [8,9]. Moreover, fusion proteins can be a result of RNA-dysregulation events like noncanonical splicing, in which fusion transcripts can be generated from *trans*-splicing [10]. In the context of immunotherapy, tumor-specific antigens resulting from gene fusions tend to be more immunogenic than classical neoantigens resulting from single nucleotide variants (SNV) or indels (insertion or deletion) [11,12].

In the last few years, there has been an increased interest in medulloblastoma-specific gene fusions. To date, there are only few gene fusions recurring in medulloblastoma (Appendix A). The *PVT1–MYC* fusion is the most frequently detected fusion that is identified in 60% of *MYC*-amplified group 3 medulloblastomas (12 out of 20 samples). Nevertheless, each detected *PVT1–MYC* fusion has different breaking points [13]. Recently, a comparison of Asian and Caucasian medulloblastoma cohorts by Luo et al. revealed 31 novel gene fusions among all medulloblastoma subgroups [14], including recurrent gene fusions such as *RAP1A–TMIGD3* with identical breaking points in two WNT MBs and one group 3 MB, *PVT1–CASC8* with different breaking points in two group 3 MBs and one group 4 MB, and three more gene fusions in at least two medulloblastoma patients affecting only the Asian cohort [14]. Furthermore, they confirmed already reported gene fusions involving *GLI2*, *DDX3X*, *SUFU* in SHH, *PVT1*, *PTEN* in group 3, and *TP53* in Group 4 [2,14]. Additionally, Jones and colleagues detected a novel fusion transcript of *DNAJB6* and *SHH* in one sample of SHH subgroup, and two further in-frame fusion transcripts in group 3 and group 4 medulloblastomas [15]. A transcriptomic analysis of 250 tumors from the SHH medulloblastoma subtype have shown an extensive network of fusions affecting *GLI2* and several loss-of function fusions involving *PTCH1*, *SUFU,* and *NCOR1* [16].

Here, we describe a new fusion transcript between the genes coding for the enhancer of polycomb homolog 2 (EPC2) and GULP PTB domain containing engulfment adaptor 1 (GULP1). The fusion is in-frame and is detected in three out of eleven medulloblastoma samples analyzed in this study irrespectively of the subtype, but not in other pediatric brain tumors entities. Importantly, the fusion was able to induce a T cell mediated immune response.

## 2. Materials and Methods

### 2.1. Patients and Material

This study was performed in agreement with the declaration of Helsinki on the use of human material for research. In accordance with the ethics committee of Rhineland-Palatinate (ethic approval no. 2021-15871), written informed consent of all patients was obtained for “scientific use of tumor tissue not needed for histopathological diagnosis” in the admission contract of the University Medical Center Mainz (§ 14 AVB). RNA of normal brain tissues (adult frontal lobe no. 110, adult parietal lobe no. 111, pons no. 168, cerebellum no. 134) was sourced from commercial vendors (Biocat, Heidelberg, Germany). Leucocyte concentrates (Buffy Coat) from healthy donors were obtained from the Transfusion Center of the University Medical Center of the Johannes Gutenberg University in Mainz.

### 2.2. Nucleic Acid Extraction

RNA extraction was performed using the RNeasy Mini Kit (Qiagen, Hilden, Germany). RNA was converted to cDNA using a PrimeScript RT reagent kit with a gDNA eraser (Takara Bio Europe, Saint-Germain-en-Laye, France). Quality control was performed using a 2100 bioanalyzer (Agilent Technologies, Waldbronn, Germany).

### 2.3. RNA-Seq

A library was constructed using the TruSeq mRNA stranded protocol (Illumina, San Diego, USA) using 2 µg total RNA. Paired-end sequencing was performed on a NextSeq500 instrument (Illumina). A total of 50 million reads (2 × 150 bp) were produced. The reads were trimmed to a maximum read length of 125 bp. TruSeq adapter sequences were trimmed. Read mapping was performed using the TopHat2 v2.0.7 aligner and the Homo sapiens UCSC hg19 reference genome (RefSeq gene annotations). Fusion calling was performed with TopHat2-Fusion v2.0.7.

### 2.4. RT-PCR and qRT-PCR

The *EPC2-GULP1* fusion gene was amplified with primers 5′-GGCAAGGACATGCCTGATCT and 5′-CCTGTATGCCAGGTCAAATGC and the circa 140 bps product was visualized on an agarose gel and analyzed by Sanger sequencing. qRT-PCR was performed with the same primers using the LightCycler 480 II Detection System and Software (Applied Biosystems, Darmstadt, Germany) with KAPA SYBR FAST LightCycler 480 Kit (PeqLab, Erlangen, Germany). After normalization to the housekeeping gene *HPRT1*, the relative quantification value was expressed as 2^−ΔΔCt^. The calibrator was calculated as the maximal number of cycles used in the PCR (40) minus the mean of the *HPRT1* Ct values, resulting in a value of 19.

### 2.5. Determination of the Full Sequence

Primers localized in exon 1 of GULP1 and exon 9, 10, 11, 12, 13 of EPC2 were used to amplify the fusion from cDNA and the products were cloned using the TA Cloning^®^ kit with pCR™2.1 Vector and One Shot^®^ TOP10F’ chemically competent E. coli (Thermo Scientific, Dreieich, Germany). Clones were analyzed by Sanger sequencing. A total of 50 ng of the DNA was used for a conventional PCR reaction using 0.1 U of Taq polymerase (Axon Labortechnik, Kaiserslautern, Germany). A total of 0.4 mM of each primer, 200 mM dNTP mix, 1.5 mM MgCl2 as well as 0.2 M betaine and the following PCR conditions: an initial denaturation step at 94 °C for 5 min, 36 cycles at 94 °C for 30 s, 58 °C for 30 s, 72 °C for 60 s, and a final extension step at 72 °C for 10 min. PCR products were purified by an enzymatic method using 10 U of exonuclease I (biolabs) and 2 U of shrimp alkaline phosphatase (SAP) for 30 min at 37 °C and 15 min at 80 °C and sequenced by StarSEQ GmbH (Mainz, Germany). The sequences were compared to the reference sequence using Sequencher program (Gene Codes, Ann Arbor, USA).

### 2.6. In Silico Epitope Prediction

In silico immunogenicity analyses were performed using the database SYFPEITHI [17] with the epitope prediction tool. The immunogenicity of epitopes, which are 9-mer peptides either spanning the fusion amino acid (AA) sequence of EPC2–GULP1 or the wildtype sequence of GULP1, was calculated for HLA allele HLA-A*0201. Furthermore, the immunogenicity of 9-mer peptides of selected, published medulloblastoma-specific fusions was also calculated for HLA haplotype HLA-A*02:01 (Caucasian population) and for HLA-A*24:02 (for fusions identified in Asian medulloblastoma cohort). Results were presented as immunoscore with the highest reachable score set at 36. A peptide was predicted as a potential immunogenic epitope if its SYFPEITHI score was ≥20 [18].

### 2.7. In Vitro Assay for Potential Peptide Antigens

PBMCs from healthy donors were isolated by Ficoll density centrifugation. Afterwards, immunomagnetic separations of CD14^+^ and CD8^+^ cells were performed with corresponding microbeads (Miltenyi Biotec, Bergisch-Gladbach, Germany). CD14^+^ cells were differentiated to immature DCs in X-VIVO15 Medium (Lonza Group, Basel, Switzerland) with 100 ng/mL IL-4 and 50 ng/mL GM-CSF (both PeproTech, Hamburg, Germany) for five days. Then, immature DCs were treated with a maturation cocktail consisting of X-VIVO15 Medium with 20 µg/mL poly I:C (Sigma-Aldrich Co., St. Louis, MO, USA), 3000 U/mL IFN-α (R&D Systems, Minneapolis, MN, USA), 1000 U/mL IFN-γ (PeproTech, Hamburg, Germany), 25 ng/mL IL-1β (PeproTech, Hamburg, Germany) and 50 ng/mL TNF-α (PeproTech, Hamburg, Germany) for 48 h. Mature DCs were incubated with peptides at a concentration of 10 µg/mL at 37 °C and 5% CO_2_ for four hours. A total of 10 µg/mL of CEF pool (JPT Peptide Technologies, Berlin, Germany) were used as positive control. Meanwhile, CD8^+^ T cells were cultured in RPMI complete medium with 20 U/mL IL-2 and 20 ng/mL IL-7 (both PeproTech, Hamburg, Germany) for seven days until co-cultivation with peptide loaded mature DCs. Co-cultivation was performed with a ratio of 10:1 (T cells:DCs). On day 20, T cells were restimulated with peptide loaded mature DCs. Cells were harvested for further analysis on day 27.

### 2.8. IFN-γ ELISpot

ELISpot analyses were performed with a human IFN-γ ELISpot kit (Autoimmun Diagnostika, Strassberg, Germany) according to the manufacturer’s instructions. For the analysis, 5 × 10^4^ cells per well were seeded. Quantification analysis was performed with AID Software V8 (Autoimmun Diagnostika, Strassberg, Germany).

### 2.9. De Novo Structure Prediction

We used the de novo structure prediction algorithm that is publicly available to inspect the possible structure of the fusion protein: The trRosetta algorithm [19]. trRosetta uses a deep-learning approach to optimize the established Rosetta algorithm. The algorithm return a pdb file, which was visualized with EzMol [20].

### 2.10. Immunohistochemistry

Immunohistochemistry was performed on 3µm thick routinely processed formalin-fixed and paraffin-embedded tissue sections. After dewaxing, antigen retrieval using EnVision FLEX target retrieval solution, with a high pH (Dako #S2368 Glostrup, Denmark) was performed. Sections were stained with anti-CD3, CD4 and CD8 primary antibody (DAKO, Glostrup, Denmark) using an immunostainer (Dako Autostainer Plus, DAKO, Glostrup, Denmark). Immunoreactivity was visualized by the universal immuno-enzyme polymer method (Nichirei Biosciences, Tokyo, Japan). Finally, sections were developed in diaminobenzidine (Lab Vision Cooperation, Fermont, CA, USA). Omission of the primary antisera in a subset of control slides resulted in no immunostaining at all.

## 3. Results

### 3.1. EPC2–GULP1 Is a Medulloblastoma Specific Fusion

We performed RNA-Seq analysis of the tumor of a medulloblastoma patient with a medulloblastoma of the SHH subtype and searched for fusion transcripts. The list of all fusions can be found in Table 1. Only the fusion between the exon 1 of EPC2 and exon 8 of GULP1 was predicted to produce an in-frame transcript (Table 1). We validated the fusion transcript by RT-PCR with specific primers and followed by Sanger sequencing, confirming the existence of the fusion and the predicted frame (Figure 1).

To assess the frequency of the fusion, we analyzed 11 medulloblastoma samples (Table 2) of different subtypes by qRT-PCR with EPC2–GULP1 specific primers. The fusion was detected in three samples (Figure 2A), with sample no. 132 being a relapse of sample no. 80. The presence of the fusion construct was confirmed by RT-PCR and Sanger sequencing of the PCR product (Figure 2B). The sequence was the same in all three samples. To assess if this fusion is medulloblastoma specific, we analyzed 24 pediatric brain tumor samples in total by RT-PCR and qRT-PCR, including the 11 medulloblastomas, eight astrocytomas, two glioblastomas, two ependymomas and one HGNET–BCOR sample (Appendix A). Furthermore, we analyzed the EPC2–GULP1 expression in four normal brain samples as control (Figure 2A). The expression of the fusion was only detected in medulloblastoma samples but not in any other brain tumor subtype or in the control samples. Interestingly, we could not find by PCR the fusion at the DNA level.

### 3.2. The Structure of the EPC2–GULP1 Fusion

Cloning of the full-length sequence indicates that the complete first exon 1 of EPC2 is included in the fusion sequence. The fusion transcript also includes the exons 8 and 9 of GULP1. Exon1 of EPC2 contains the N-terminal enhancer of polycomb-like domain A (EPcA), which is also found at the N-terminal of the EPL1 protein family, who are a member of a histone acetyltransferase complex. These complexes are involved in transcriptional activation of selected genes [21]. EPC2 has 14 exons in total. GULP1 is keeping 2 of its 13 coding exons. GULP1 has a phosphotyrosine-binding (PTB) domain followed by a coiled-coil domain. While only a short part of PTB is included in the fusion construct, the coiled-coil domain is completely present. The fusion protein and its domains are shown in Figure 3A. To analyze if the structure of the coiled-coil domain is maintained, we used a structure prediction algorithm. The helix of GULP1s coiled-coil domain is predicted by trRosetta, which also predict the amino acids of EPC2 to be connected to the helix with a turn. Figure 3B shows trRosettas result.

### 3.3. EPC2–GULP1 Fusion Peptide Has Immunogenic Potential to Activate CD8^+^ T Cells

It is known that the associated peptides of tumor specific mutations, such as fusion peptides resulting from translocations, might be immunogenic [22]. Therefore, we performed in silico epitope prediction of EPC2–GULP1 peptides and other fusions described in the literature for MHC class I type HLA-A*02:01 (Caucasian population) or with HLA-A*24:02 (for fusions described in the Asian population) with the SYFPEITHI algorithm (Ver. 1.0) [17]. A total of 50% of the analyzed fusions carried a peptide with a immunoscore of at least 20, including the 9-mer AA-sequence SAEEITLTI of the EPC2–GULP1 fusion (see Appendix A).

The immunogenic potential of the medulloblastoma specific EPC2–GULP1 fusion peptide was tested on CD8^+^ T cells from three healthy donors with HLA-A*02:01 specification towards the IFN-γ secretion. As negative control, we used the 9-mer wildtype sequence of GULP1 RKFLESGGK with an immunoscore of 2. Figure 4 shows the results of the in vitro assay of healthy donor one that had the strongest reactivity of CD8^+^ T cells towards the fusion peptide compared to the wildtype peptide.

Moreover, CD8^+^ T cells of healthy donor two and three showed also a higher reactivity towards the fusion peptide, whereas the T cells of healthy donor two showed also a strong reactivity against the DMSO vehicle control (Appendix A). The quantitative results of all three samples are summarized in Table 3.

Interestingly, sample no. 25 carrying the fusion shows an infiltration of CD3^+^ T cells and particularly of CD8^+^ cells, while no infiltration was observed in a fusion-negative sample (Appendix A, sample no. 129).

In conclusion, our work indicates that the medulloblastoma specific fusion peptide SAEEITLTI mediates a T cell specific immune response.

## 4. Discussion

Medulloblastoma is the most frequently diagnosed embryonal tumor of the CNS in children. So far, therapeutic strategies are limited to radiation, chemotherapy, and surgery with severe neurological side effects. Thus, additional strategies such as targeted therapies and immunotherapy are currently under investigation. However, pediatric medulloblastoma is considered a tumor entity with minimal mutational load [23] and, therefore, low immunogenicity. Here, we identified a new fusion transcript potentially defining a more immunogenic subgroup of medulloblastoma.

*EPC2* and its paralog *EPC1* were first described by Stankunas et al. in 1998 as a member of the polycomb group of genes (PcG). They reported that EPC has repressive characteristics by being part of a chromatin regulatory mechanism [24]. Later on, it was shown that the EPC proteins have not only repressive but activating capabilities as well: the first domain EPcA is linked to a zinc-finger domain, indicating that EPC2 could interact with DNA. It functions as a strong transcriptional activator and it is suggested that EPC2 has activating as well as repressing function, depending on its splice variant [25]. The EPcA domain is also important for nucleosomal histone acetyltransferase (HAT) activity [26]. The EPcB and EPcC domains reportedly have repressive characteristics. The EPcC domain is involved in heterochromatin formation [27,28]. Thus, EPC2 has oncogenic potential since it has an impact on transcription and as part of a regulatory complex contributes to cellular processes such as induction of apoptosis [29]. EPC2 and EPC1 were identified as critical oncogenic cofactors in AML as part of the EP400 complex. Both proteins were also shown to sustain oncogenic potential in MLL leukemia stem cells [30]. Contrary, Gotoh et al. showed that reduced expression of EPC2 was correlated to tumor aggressiveness and, therefore, may play a role in malignant progression [31]. These seemingly contrary findings could be explained with the already mentioned dual transcriptional activity of EPC2 [25]. Depending on its splice variant, it has different transcriptional activities. Thus, in a fusion event, the retained and lost domains are pivotal to predict pathogenic impact of the fusion protein. The fusion protein described in this work has only a small portion of the normal EPC2 protein. The EPcB and EPcC domains are lost completely and with them the transcriptional repressive function of EPC2. However, the EPcA domain is retained, except for its zinc-finger portion. Therefore, the DNA binding capability is lost, but not necessarily decreasing EPcAs function to zero.

GULP1 is a nucleocytoplasmic shuttling protein that is involved in engulfing apoptotic cell debris and lipid homeostasis [32]. Mediated by low density lipoprotein receptor-related protein 1 (LRP1), GULP1 shows transactivational activity of liposaccharide cholinephosphotransferase (LICD) [33]. In ovarian cells, GULP1 regulates TGF-β response, and it is required to maintain their sensitivity to cell growth arrest. If the protein is lost in ovarian cells, it contributes to ovarian cancer progression [34]. GULP1 is considered to be a tumor suppressor gene, showing anti tumoral function not only in ovarian cells but also in urothelial carcinoma [35,36]. In the fusion construct, GULP1 maintains the majority of its coiled-coil domain and could thus still be interacting with DNA or other proteins. The coiled-coil region of GULP1 is indeed predicted to mediate dimers formation, facilitating the formation of such complexes [37].

Alongside the creation of new functionally active proteins, fusions may generate neoantigens that are recognized by the immune system. In this case, an amino acid sequence spanning the fusion points of two genes differs from the respective wild-type sequences, so that a tumor-specific immune response is possible [38]. Multi-peptide vaccination with fusion peptide spanning the BCR–ABL fusion showed a specific immune response and improved disease control in patients with chronic myeloid leukemia [39,40]. Moreover, gene fusions and their peptides in prostate cancer are also classified as immunogenic [38]. Chang and colleagues analyzed somatic alterations and gene fusions of 23 pediatric tumor entities and could identify potential neoepitopes based on missense mutations and resulting from gene fusions [41]. Interestingly, gene fusions are associated with a better response to immunotherapy in melanoma [42]. Furthermore, Yang et al. have studied a cohort of head and neck tumors with low mutational burden and minimal immune infiltration, and identified gene-fusion derived neoantigens, e.g., the in-frame *DEK–AFF2* gene fusion, that generate cytotoxic T cell responses [22]. A 9-mer peptide from the CBFB–MYH11 fusion protein in acute myeloid leukemia (AML) enables CD8^+^ T cells to kill AML cell lines [43]. Taken together, fusion transcripts are emerging targets for immunotherapies with uses for the development of tumor vaccines and adoptive cell therapies. Recent studies have shown that fusion transcripts seem to have a higher immunogenic potential than SNV and indel-based candidate neoantigens and, therefore, they may be better candidates for cancer vaccines [11].

Given the fact that high SNV or indel mutational burdens are limited, especially in pediatric malignancies, the repertoire of tumor antigens needs to be broadened [12]. Several clinical studies are currently evaluating immunotherapy in medulloblastoma [44]. Cytotoxic T cells can infiltrate medulloblastoma [45] and checkpoint inhibitors against PD-1 are under investigation in clinical trials recruiting medulloblastoma patients and other CNS tumors. Although immunotherapeutic approaches for medulloblastoma patients such as cancer vaccines, natural killer cells and CAR T cells are promising [44], one of the major limitations in treating medulloblastoma with immunotherapy is the low immunogenicity and mutational load. It is interesting to note that few reported medulloblastoma-specific fusion peptides, such as MLLT6–MRPL45, LCLAT1–ERBB4, ASAP1–WD4HV1 and PTEN–THAP9 [2,14,15] have similar immunogenic potential to the EPC2–GULP1 fusion peptide SAEEITLTI. Therefore, future studies are needed to determine whether medulloblastoma patients with *EPC2-GULP1* or other fusions have a higher response rate to immunotherapy.

Interestingly, the *EPC2–GULP1* fusion was detected in one patient with WNT and one patient with SHH MB subtype, and thus, the fusion is not associated with a particular subtype. Tumor classification based on molecular profiling is improving disease management of medulloblastoma and other tumors but is generally not taking into account the tumor microenvironment and the immune landscape. However, recent works suggest the existence of tumor subtypes based on tumor immune signatures, helping guide immunotherapy or prognostic prediction [46,47,48]. Thus, defining the immunogenicity of medulloblastoma may help to identify subsets with potential for immune responsiveness. The presence of immunogenic fusions could be a factor helping in the definition of such subtypes in medulloblastoma and other tumor entities. Recently, Luo et al. identified five different gene fusions affecting the WNT MB subgroup. Only two of them (*RAP1A–TMIGD3* and *ZSWIM5P2–MEIS3*) occurred in both the WNT and SHH subtype as well as in group 3 and group 4 medulloblastoma [14]. Based on these findings, it is worth exploring whether the *EPC2–GULP1* fusion might also appear in group 3 and group 4 medulloblastoma. For that, however, analyzing a larger cohort of patients would be indispensable.

The fusion we report in this work has not been described in other works analyzing the transcriptional landscape of medulloblastoma so far [14,16]. The landscape of gene fusions in medulloblastoma is very heterogeneous with a low recurrence of the same fusion in different patients. If the algorithm used for fusion calling (TopHat-Fusion in this work vs. Arriba and STAR-fusion, InFusion, Trans-Abyss in [16]) may have influenced the detection of the fusion, remains to be elucidated. Because we validated the presence of the fusion by qRT-PCR and sequencing using specific primers, an artifact of the fusion detection algorithm can be excluded.

Fusion transcripts do not only arise from translocated chromosomes but can also be produced via *trans*-splicing, which occurs when two transcripts from different origin are spliced together during mRNA processing [49]. Since we could not find a chromosomal fusion with the DNA motif of the reported fusion but found the fusion transcript in multiple samples, we assume that the fusion transcript *EPC2–GULP1* is derived from *trans*-splicing, even if this assumption should be validated by more specific methods, as for example FISH or whole exome sequencing (WES). *EPC2* is already a gene edited with alternative splicing between its long and short variant. It would be interesting to investigate if the spliceosome is aberrant in these samples, and that led to the *trans*-splicing event.

## 5. Conclusions

High SNV or indel mutational burdens are generally very limited in pediatric malignancies, e.g., in medulloblastoma. Therefore, gene fusions are promising targets for these tumors, as they are described to be potentially highly immunogenic. The *EPC2–GULP1* fusion transcript is medulloblastoma-specific occurring in the SHH and WNT medulloblastoma subgroup. Further studies with a larger patient cohort are required to elucidate whether the *EPC2–GULP1* fusion might also appear in group 3 or group 4 medulloblastoma. Our data indicate a T cell mediated immune response in form of IFN-γ secretion. Taken together, these aspects support an immunotherapeutic approach for pediatric medulloblastoma patients carrying the *EPC2–GULP1* fusion and possibly other gene fusions and could contribute to an immunogenicity-based stratification of medulloblastoma.

## Figures and Tables

**Figure 1 cancers-13-05838-f001:**
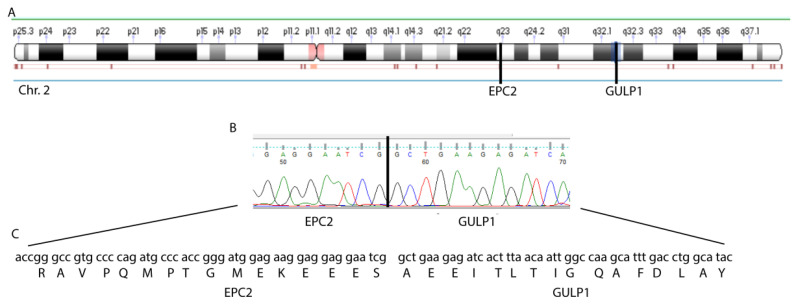
Detection of the EPC2–GULP1 Fusion. (**A**) Position of EPC2 and GULP1 on chr.2. (**B**) Sanger sequencing showing the sequence of the fusion position. (**C**) Protein sequence of the fusion transcript predicted by NGS and validated by Sanger sequencing.

**Figure 2 cancers-13-05838-f002:**
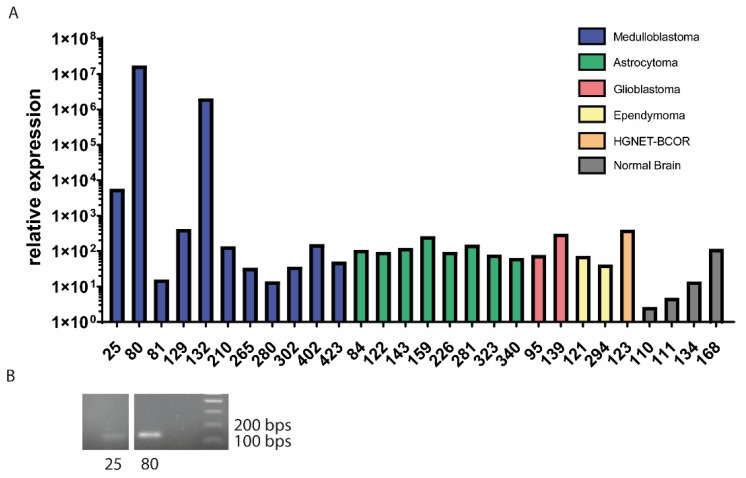
The EPC2–GULP1 fusion is selectively expressed in medulloblastoma samples. (**A**) The expression of the fusion was analyzed by qRT-PCR in the indicated samples. (**B**) The RT-PCR product of two positive samples was analyzed on a gel and was used for Sanger sequencing. The size was the same in both samples.

**Figure 3 cancers-13-05838-f003:**
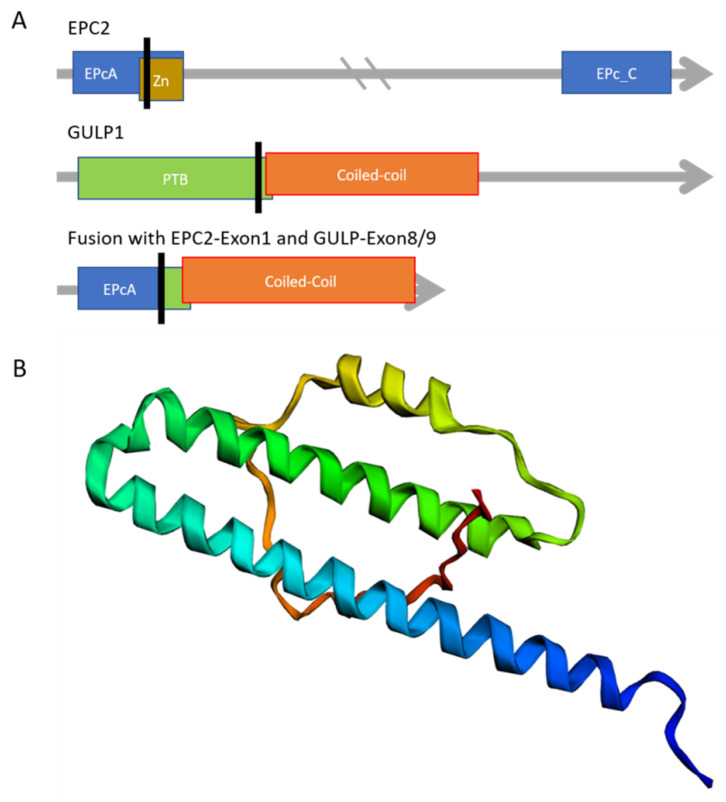
Possible composition of the EPC2–GULP1 fusion protein. (**A**) The fusion protein contains the 51 first amino acids of EPC2 and, therefore, the main part of the EPL1 domain, excluding the zinc finger domain (Zn). Known active domains of GULP1 are the phosphotyrosine binding (PTB) and a coiled-coil. The black line represents the breaking points and the position of fusion in the transcripts or proteins. (**B**) Structure of the fusion protein predicted by trRosetta. The α-helix represents the coiled-coil domain of GULP1. The TM-score had an estimated value of 0.264.

**Figure 4 cancers-13-05838-f004:**
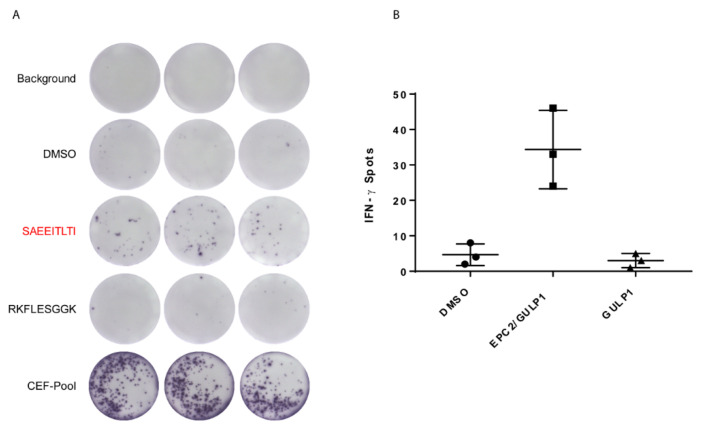
CD8^+^ T cells are reactive towards fusion peptide. (**A**) ELISpot analysis after in vitro assay based on co-culturing peptide loaded DCs with CD8^+^ T cells of healthy donor no. 1. DMSO was taken as vehicle control, wildtype peptide of GULP1 as negative control and CEF-pool as positive control. The fusion peptide is marked red. Quantification of the spots is shown in (**B**). Mean values and standard abbrevation of IFN-γ spots after stimulation with the fusion peptide and wildtype peptide. DMSO treated cells were used as negative control.

**Table 1 cancers-13-05838-t001:** Fusions detected in Medullobalstoma sample no. 80.

Transcript 1	Chr	Position	Exon	Transcript 2	Chr	Position	Exon	In-Frame
*KPNA3*	chr13	50321083–50321128	exon 16	*TNFSF13B*	chr13	108955599–108955712	exon 4	no
*KPNA3*	chr13	50321083–50321128	exon 16	*TNFSF13B*	chr13	108939207–108955598	intron 3	no
*EPC2*	chr2	149402558–149402738	exon 1	*GULP1*	chr2	189433964–189434081	exon 8	yes
*EPC2*	chr2	149402739–149447780	intron 1	*GULP1*	chr2	189433964–189434081	exon 8	no
*NFIB*	chr9	14306986–14307518	exon 8	*ENSG00000237137*	chr9	14531912–14532039	exon 1	no

**Table 2 cancers-13-05838-t002:** Overview of medulloblastoma samples analyzed in this work.

Sample	Age at Diagnosis	Genetic Subtype	Histologic Subtype	*EPC2–GULP1* Fusion
25	16	WNT	desmoplastic medulloblastoma	yes
80	6	SHH	large cell/anaplastic medulloblastoma	yes
81	5	Group 3/4	classic medulloblastoma	no
129	8	WNT	classic medulloblastoma	no
132 *	7	SHH	large cell/anaplastic medulloblastoma	yes
210	5	Group 3/4	classic medulloblastoma	no
265	11	Group 3/4	large cell/anaplastic medulloblastoma	no
280	6	Group 3	classic medulloblastoma	no
302	1	SHH	desmoplastic medulloblastoma	no
402	1	SHH	desmoplastic nodular medulloblastoma	no
423	6	Group 3/4	classic medulloblastoma	no

* Same patient as sample no. 80: no. 132 is the relapse of no. 80.

**Table 3 cancers-13-05838-t003:** IFN-γ secretion after stimulation with the fusion peptide.

Healthy Donor	IFN-γ Secretion of CD8^+^ T Cells
1	1144%
2	131%
3	354%

The spot number of the wildtype peptide was defined as 100%.

## Data Availability

The data presented in this study are available on request from the corresponding author.

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
