# Peer review of "Identification of an Immunogenic Medulloblastoma-Specific Fusion Involving EPC2 and GULP1"

_cancers, 2021, doi:10.3390/cancers13225838_

Round 1
Reviewer 1 Report
In this article Paret et al after reviewing fusions in human medulloblastoma and analyzing their predicted immunogenicity, described a new fusion protein (EPC2-GULP1) identified by RNA-Seq. This fusion was presented in samples from other pediatric brain tumors. The Authors also analyzed the ability of EPC2-GULP1-derived peptide to bind to HLA-A*0201 in silico. They also showed that this peptide was inducing IFN-g production by CD8 T cells in vitro, suggesting that immunotherapeutic approaches might be beneficial in patients carrying this fusion (or the other fusions analyzed in the first part of the article).
Overall, the manuscript is of interest.
I have some minor points to highlight.
English usage is generally correct but there are some specific points that should be checked. For instance on page 1, lines 35-36 (In our study, we identified a new medulloblastoma-specific fusion transcript EPC2-GULP1, which was able to activate CD8+ T cells). The transcript per se is not able to activate CD8 cells as the use of “which” would indicate in that sentence (but it is the protein the transcript codes for to stimulate CD8 cells).
In the abstract and later on in the text the Authors wrote that 3 out 11 samples were carrying the EPC2-GULP1 fusion. Even if this is correct, it might induce the reader to think that about one fourth of the samples were positive. It would be better to write also in the abstract that two of these samples were from one patient (the second sample being from a relapsed tumor, as written later on in the text).
How do the Authors explain that T cells from the healthy donor number two produced IFNg in response to DMSO (vehicle control; Fig. S1)? Did the assay have any technical issue?
Results shown in Fig 4 and in Fig. S1 could be joined, eventually together with those from Table 4, in one more comprehensive figure 4.
Author Response
English usage is generally correct but there are some specific points that should be checked. For instance on page 1, lines 35-36 (In our study, we identified a new medulloblastoma-specific fusion transcript EPC2-GULP1, which was able to activate CD8+ T cells). The transcript per se is not able to activate CD8 cells as the use of “which” would indicate in that sentence (but it is the protein the transcript codes for to stimulate CD8 cells).
We thank you for the helpful hint. In the revised manuscript, we corrected the sentence in line 36-37 „The resulting protein sequence produced a neoantigen, which was able to activate CD8+ T cells.“
In the abstract and later on in the text the Authors wrote that 3 out 11 samples were carrying the EPC2-GULP1 fusion. Even if this is correct, it might induce the reader to think that about one fourth of the samples were positive. It would be better to write also in the abstract that two of these samples were from one patient (the second sample being from a relapsed tumor, as written later on in the text).
Thank you for this advice. We have added the information about the patients in the abstract at line 48-49 : “By qRT-PCR analysis, the fusion was detected in 3 out of 11 medulloblastoma samples, whereby 2 samples were from the same patients obtained at two different time points (initial diagnosis and relapse), but not in other pediatric brain tumor entities.“
How do the Authors explain that T cells from the healthy donor number two produced IFNg in response to DMSO (vehicle control; Fig. S1)? Did the assay have any technical issue?
Generally, background from DMSO is not expected in Elispot assay at the low concentration (0.01%) used in our assay, but we can not exclude a batch and patient specific effect. Indeed, an induction of IFNg release via DMSO has been already reported by Gamage et al (PMID 28827286) and DMSO is known to have different off-target effects on cells (PMID 33802212). Such effects could be influenced by the individual physiological characteristics of donor 2 who had an overall stronger reaction in term of IFNg production compared to donor 1 and 2. The DMSO dilution used for vehicle control was always freshly prepared, while, the peptides were diluted at the same time with sterile H2O and the same batch of DMSO. Therefore, we think that a direct comparison between the fusion and wildtype peptide in term of IFNg release by donor 2 is still possible.
Results shown in Fig 4 and in Fig. S1 could be joined, eventually together with those from Table 4, in one more comprehensive figure 4.
We are very thankful for the idea, but we decided not to change the figure because, in our opinion, the current figure has a better clearness than a more comprehensive version by joining Fig. 4 and Fig. S1.
Reviewer 2 Report
In this work the authors describe the fusion of EPC2 and GULP1 in patients with medulloblastoma.
The introduction is adequate and the methodological approach is solid. The results are presented in a clear and truthful way. The conclusions reached by the author are supported by his findings.
Minor concern:
The fusion was detected in 3/11 cases. This finding could allow for an immunogenic subset of medulloblastomas. The ultimate goal would be to make it a therapeutic target, but on what do the authors base themselves to think that their findings may contribute to the already established subgroups of medulloblastomas?
Author Response
The fusion was detected in 3/11 cases. This finding could allow for an immunogenic subset of medulloblastomas. The ultimate goal would be to make it a therapeutic target, but on what do the authors base themselves to think that their findings may contribute to the already established subgroups of medulloblastomas?
This is indeed an interesting point for the discussion. We know from studies in other tumor entities that tumor subtypes based on tumor immune signatures exist, which may help to guide immunotherapy or prognostic prediction We added this aspect in the discussion (line 410 ff.) and in the conclusions (line 454f.)
Discussion:
Tumor classification based on molecular profiling is improving disease management of medulloblastoma and other tumors, but is generally not taking in account of the tumor microenvironment and the immune landscape. However, recent works suggest the existence of tumor subtypes based on tumor immune signatures, helping guide immunotherapy or prognostic prediction [46-48]. Thus, defining the immunogenicity of medulloblastoma may help to identify subsets with potential for immune responsiveness. The presence of immunogenic fusions could be a factor helping in the definition of such subtypes in medulloblastoma and other tumor entities.
Conclusions:
Taken together, these aspects support an immunotherapeutic approach for pediatric medulloblastoma patients carrying the EPC2-GULP1 fusion and possibly other gene fusions and could contribute to an immunogenicity-based stratification of medulloblastoma.